# Evaluation of Snowmelt Impacts on Flood Flows Based on Remote Sensing Using SRM Model

Mohammad Reza Goodarzi [1,*], Maryam Sabaghzadeh [1] and Majid Niazkar [2,*]

[1] Department of Civil Engineering, Yazd University, Yazd 8915813135, Iran
[2] Faculty of Science and Technology, Free University of Bozen-Bolzano, Piazza Università 5, 39100 Bolzano, Italy
* Correspondence: goodarzimr@yazd.ac.ir (M.R.G.); majid.niazkar@unibz.it (M.N.)

**Abstract:** Snowmelt is an important source of stream flows in mountainous areas. This study investigated the impact of snowmelt on flooding. First, the study area was divided into four zones based on elevation. Second, the Snow-Covered Area (SCA) from 2013 to 2018 was estimated from daily MODIS images with the help of Google Earth Engine. Runoff in the area was then simulated using the Snowmelt Runoff Model (SRM). As a result, short periods with high runoff and the possibility of floods were identified, while the contribution of snowmelt and rainfall in the total runoff was separated. The results showed that while the snowmelt on average accounted for only 23% of total runoff in the zone with elevation under 2000 m, the ratio increased with elevation, ultimately reaching as high as 87% in the zone with elevation above 3000 m. As the height increases, the effect of snow on runoff and flooding increases so much that it should not be ignored. However, in most hydrological studies, the effect of snow is ignored due to the lack of sufficient data about snow. This study showed that snow can be very effective, especially in high areas.

**Keywords:** flood; MODIS; remote sensing; snow; SRM

## 1. Introduction

Covering approximately 40–50% of the northern hemisphere in winter, snow is a major determinant of the water balance in many parts of the world [1]. Snowfall tends to take place at high elevations in poorly accessible or inaccessible areas where it is difficult to build and maintain meteorological stations. Thus, satellite imagery has become a very popular tool for studying snowmelt-dominated basins since its advent in the 1960s [2]. With rising concerns about early snowmelt due to global warming, more research has been conducted on how snowmelt timing can change under different climate change scenarios [3–5]. These scenarios can also be used to predict the future of floods [6].

Satellite images can be used to determine the snow-covered area (SCA) and the corresponding snow-water equivalent (SWE) [7]. Snowmelt tends to occur in the warmer months of spring and summer. In spring, this may coincide with intense rainfalls that generate runoffs exceeding the soil infiltration capacity, leading to flooding. Rain on snow events also lead to flooding. Spring snowmelt can indeed affect the volume of these floods.

A variety of models can be used to estimate the snowmelt runoff and its contribution to floods so that the necessary warnings can be issued in due time to prevent loss of life and property [8,9]. In this study, the Snowmelt Runoff Model (SRM) is used for this purpose. This model simulates and predicts daily flows in mountainous catchments and other types of catchments where snowmelt constitutes a major part of the runoff. The SRM model has been used in numerous studies to predict snowmelt runoffs. In one of these studies, Meng et al. [10] used daily MODIS images with a spatial resolution of 500 m to estimate the amount of runoff generated due to the snowmelt from 2003 to 2009 in a basin in China. The results showed that the highest monthly runoff belonged to July and the lowest belonged to October. The total runoff was found to have an increasing trend from May to October. In

the study evaluations, the coefficient of determination was estimated to be 0.73 with an 8.85% volume difference, showing good accuracy.

According to the literature, only a few studies have considered the effect of snow on flooding. In one of these studies, Graybeal and Leathers [11] examined the risk of flooding due to the snowmelt in the Appalachian Mountains in the United States. In the study, flooding was a function of the amount of snowfall and snow depth, which were used to obtain a regional pattern for the period from 1971 to 2000. The Gumbel distribution was then used to map the seasonal total snowfall and seasonal maximum snow depth. However, they did not make clear how much snow or rain affected each flood. Zakharova et al. [12] also investigated the role of snowmelt in spring floods. For this purpose, they used the snow depth estimates obtained by microwave measurements to forecast the volume of water stored in winter for the period from 1989 to 2006. They estimated that 30% of the water from snowmelt will evaporate or penetrate the ground rather than runoff. In another study, Nester et al. [13] used a semi-distributed hydrological model to develop a rainfall-runoff model for the upper Danube basin. This model, which is similar to the HBV model for snow, was used to determine the snow-water equivalent for the years 2003 to 2009. The SCA was also estimated using MODIS images, while they merely discussed the estimation of snow cover and runoff. Furthermore, Qiao et al. [14] utilized the Geographic Information System (GIS) and remote sensing to analyze the factors that influence snowmelt-induced floods, such as the SCA, snow depth, air temperature, precipitation, topography, and land cover. They used their results to develop a model for predicting and assessing flood damage. Their model can predict the flood area, depth, and how much damage it will cause in each area. In another study, Uwamahoro et al. [15] exploited an algorithm operating based on temperature to decompose snow and rainfall-runoffs in two catchments. They employed the Soil and Water Assessment Tool (SWAT) model to estimate the volume of flooding and snowmelt in the area. Then, they modified the model according to the conditions and compared its results with those of the original one. In general, the modified model was found to provide more accurate results in terms of the snow contribution to runoff, peak flood, and flood frequency.

In the present study, the goal was to investigate the effect of snowmelt on river flow and ultimately flooding. Most studies conducted in this field have concentrated more on modeling snowmelt runoff rather than the impact of snowmelt on flooding. Given the time needed to download and process satellite images, previous studies have mostly limited themselves to short periods. However, in this study, Google Earth Engine was used to perform the analyses for an extended period. The SRM was then applied to model the snowmelt runoff. Given the impact of elevation on temperature and snowfall, the study area was divided into four altitude zones with a 500 m distance. Then, the effect of the snowmelt runoff on flooding was examined for each zone separately. The findings of this study can facilitate the assessment of flood risk in the studied region and assist local decision-makers in planning and preparing to deal with such natural threats.

## 2. Materials and Methods

### 2.1. Study Area

The middle Ajichai catchment is part of the Ajichai basin and one of the most important sub-basins of Lake Urmia in northwestern Iran. Bounded between 36°47′ and 39°23′ northern latitudes and 45°05′ and 48°18′ eastern longitudes, this catchment has an area of about 5676 km$^2$. It starts from the southern and southwestern slopes of Sabalan Mountain about 33 km northeast of the city of Sarab at an elevation of 3400 m, passes north of the city of Tabriz, and ends in Lake Urmia at an elevation of 1270 m. In terms of topography, the catchment has a minimum height of 1522 and a maximum height of 3656 m. According to observations, the average annual rainfall is 350 mm. The highest amount of rain is in the months of March and April with 20% rainfall throughout the year, which increases the probability of flooding. The lowest amount corresponds to the months of July and August with 0.1% precipitation throughout the year. The highest amount of snow is in January and

the summer months are without snow. The average annual temperature is about 7.8 °C in the basin. The absolute maximum annual temperature is 30 °C in July and the absolute minimum temperature is −15.8 °C in February. The average flow rate of the AjiChai River is about 5 m³/s annually, which is highest in April and May, and about 15 and 6 m³/s, respectively. Due to the low rainfall in spring and the delay in the snow melting, it is expected that snow melting will play a significant role in the runoff of the river.

The location of the study area is displayed in Figure 1. In this study, we used daily precipitation and temperature data recorded at meteorological stations and also the daily runoff data recorded at Merkid station to validate the simulation results. The studied area includes the three cities of Sarab, Haris, and Bostan Abad, where floods have been reported. Among the floods, we considered the ones that occurred on 23 June 2018, 12 April 2015, and 14 April 2017, which were associated with human and financial losses. The location of the mentioned stations in the catchment is marked in Figure 1. More detailed information about these stations is provided in Table 1.

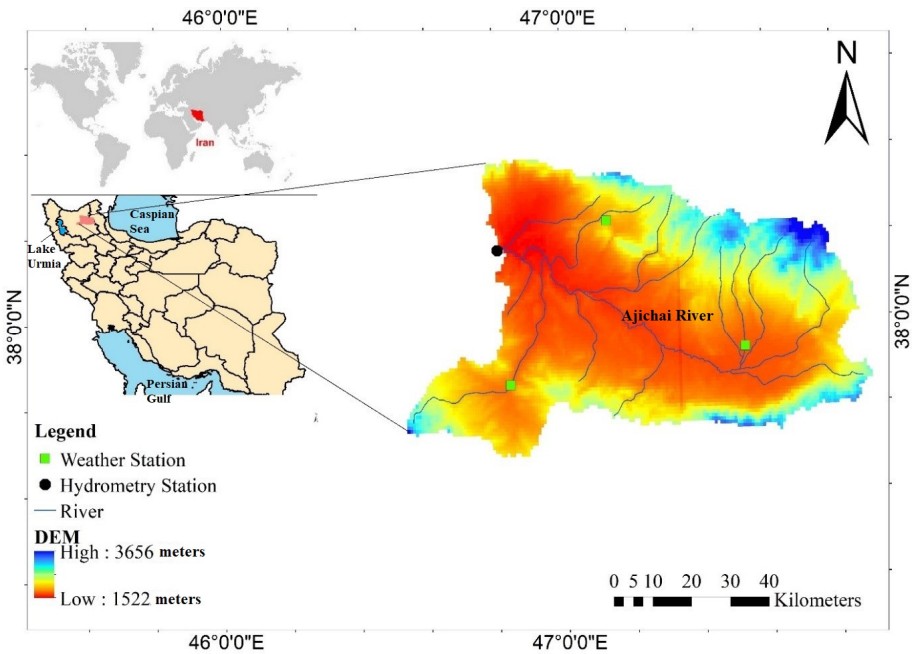

**Figure 1.** Location of the case study area.

**Table 1.** Details of data products in this study.

| Satellite Products | | | | |
|---|---|---|---|---|
| Data product | Band name | Resolution | Data period | Data |
| MOD10A1.006 Terra Snow Cover Daily Global 500 m | NDSI_Snow_Cover | 500 m | 2013–2018 | Snow cover |
| **Ground station observation** | | | | |
| Station name | Elevation | Longitude | Latitude | Zone |
| Heris | 1950 | 47°7′48″ | 38°13′47″ | A |
| Sarab | 1682 | 47°31′48″ | 37°55′47″ | A |
| Bostanabad | 1736 | 37°51′0″ | 46°50′24″ | A |
| Merkid | 1532 | 46°47′59″ | 38°9′35″ | Zone |

*2.2. Methodology*

In this study, the runoff generated from the snowmelt was modeled using the SRM. To use this model, the study area was first divided into four zones based on elevation with a

500 m distance. The reason for this division is the effect of altitude on the amount of snow in the area, as areas with higher altitudes usually have more snow, and consequently the simulation accuracy increases with this division. For each altitude zone, the input data were entered separately, and the simulation was conducted separately.

These four zones are shown in Figure 2a and their characteristics are given in Table 2. As shown in the table, the largest zone is Zone A (elevation range of 1522–2000 m) with an area of 3563 km², which covers 62.89% of the study area including the cities of Haris, Sarab, and Bostanabad. The smallest zone is Zone D (elevation range of 3000–3656 m) with an area of 77 km², which constitutes only 1.36% of the total area of the catchment but comprises the points with the highest elevation in the area, including the mountains of Sabalan and Sahand, which are covered with snow most of the year. The areas of these four zones are compared in Figure 2b.

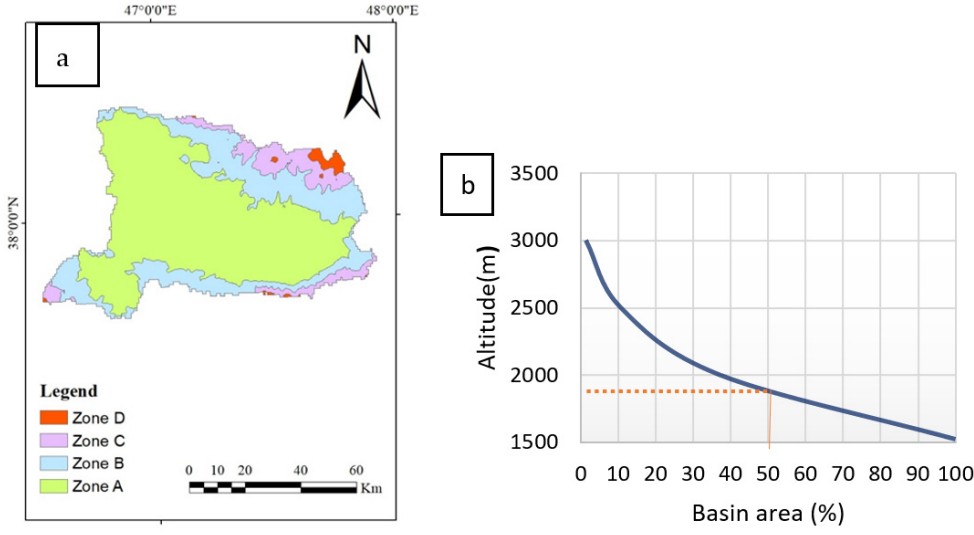

**Figure 2.** (**a**) Zonal divisions of the basin (**b**) Hypsometric chart.

**Table 2.** Elevation zone information.

| Zone | Elevation Range (m) | Hypsometric Average Elevation (m) | Area (km²) | Area Percent (%) |
|---|---|---|---|---|
| A | 1522–2000 | 1761 | 3563 | 62.89 |
| B | 2000–2500 | 2250 | 1498 | 26.44 |
| C | 2500–3000 | 2750 | 527 | 9.3 |
| D | 3000–3656 | 3328 | 77 | 1.36 |

The required input data of the model is explained in detail in Section 2.3.2. Average daily temperature and precipitation data were taken from the meteorological stations, whose specifications are given in Table 1, and given as input to the model. Since all stations are in the first altitude zone, to obtain the temperature and precipitation of other altitude zones, 3% was added to the precipitation for every 100 m of altitude increase and 0.4 °C was reduced from the temperature. The ratio of the snow-covered area is another important input that can be calculated using satellite images obtained for each area separately, then divided by the total area of the altitude area and entered into the model. The daily runoff data were taken from the hydrometric station, whose specifications are given in Table 1, and given to the model. This model needs parameters, which are explained in Section 2.3.3. Due to the large number of parameters, the more effective parameters were first determined by performing sensitivity analysis, and then using the values obtained from similar works, the allowed intervals were determined for each parameter by the model and the correct and error values of the parameters were determined, and the model was executed. Then, the annual hydrograph of the studied area was simulated by the model, and according to the

observed runoff values, the accuracy of the modeling was calculated using the coefficient of determination and the volume difference coefficient. After the implementation of the SRM model, it gave the runoff caused by snow melting in each altitude area separately. In this study, in order to determine the effect of snow on floods, short periods of time with a significant runoff were separated in each altitude area and considered as times with the possibility of flooding. The effect of snow and rain was separated in each area, and the results obtained will be analyzed in the results section.

### 2.2.1. SCA Estimation by Remote Sensing

MODIS was the first sensor whose products were used for SCA mapping [16]. The National Snow and Ice Data Center (NSIDC) introduced MODIS products for generating daily, eight-day, and monthly SCA maps. Using these products, the Normalized Difference Snow Index (NDSI) can be derived from visible and near-infrared channels using Equation (1) [17].

$$NDSI = (b4 + b6)/(b4 - b6) \tag{1}$$

Here, $b4$ denotes Band 4 of MODIS with a wavelength range of 0.54–0.56 μm, and $b6$ denotes Band 6 of MODIS with a wavelength range of 1.62–1.65 μm.

Having near-zero reflectance, snow can be distinguished from clouds using NDSI. While NDSI > 0.4 normally indicates snow cover, it only applies to places where the surface is smooth. In forested areas, the vegetation covering the snow makes it more difficult to distinguish snow-covered areas. According to the research conducted by Hall et al. [18], NDSI > 0.4 is only accurate in places where the Normalized Difference Vegetation Index (NDVI) is around 0.1. Basically, while the daily snow cover product MOD10A1 is about 93% accurate if the sky is clear, it also depends on vegetation cover [19]. Under general climatic conditions, the accuracy is about 31% for the Aqua satellite (MYD10A1) and about 45% for the Terra satellite (MOD10A1) [20]. Using the NDSI index, the snow pixels were identified and then the snow-covered area was calculated by dividing the snow pixels by the total number of pixels in the area.

### 2.2.2. Google Earth Engine

Google Earth Engine is a web-based service first released in April 2008 whereby a wide variety of remote sensing processes can be executed on a massive satellite imagery database using simple commands [21,22]. In general, the process of remote sensing images for research purposes involves downloading images of interest, applying geometric and radiometric corrections, and then applying the required processes, which will vary depending on the research subject. Each of these steps has its own challenges. For example, given the large size of satellite images, it could take a long time to download them, and they tend to take up a lot of space and are difficult to handle. Another problem is that the mentioned corrections cannot be made without powerful software. Therefore, it could become difficult and tedious to work with satellite images for long periods. With the advent of Google Earth Engine, however, researchers can quickly process nearly 5000 satellite images free of charge [23]. In this study, Google Earth Engine was used to estimate the daily snow cover of the study area from 2013 to 2018.

### *2.3. SRM*

### 2.3.1. Model Structure

The SRM has been designed to simulate and predict daily flows in mountainous catchments [24]. This model can also apply and examine the impacts of climate change on snowmelt runoff. It was first proposed by Martinec in 1975 [25]. With the growing use of satellite imagery for remote sensing purposes, including the estimation of snow-covered areas, SRM has also been used for larger basins. The largest basin to which this model has been applied so far is about 918,144 km$^2$ and has an elevation range of 8840 m. The SRM does not seem to have any limitations in terms of area and elevation range [26]. To evaluate

its accuracy, the daily flow estimates of the model must be compared with runoff measured in the area.

In SRM, the runoff generated from the snowmelt and rainfall is estimated daily using Equation (2) [25].

$$Q_{n+1} = [C_{sn}\, a_n\, (T_n + \Delta T_n)\, S_n + C_{Rn}\, P_n]\, (A \cdot 10{,}000)/86{,}400\, (1 - k_{n+1}) + Q_n\, k_{n+1} \qquad (2)$$

where:

$Q$ = mean daily discharge (in $m^3\ s^{-1}$)

$C_s$ = snowmelt runoff coefficient,

$C_R$ = rainfall-runoff coefficient

$a$ = degree-day factor (in $cm.°C^{-1}\ d^{-1}$), which indicates the snowmelt depth due to 1 degree-day

$T$ = the number of degree-days (in °C d)

$\Delta T$ = temperature adjustment based on the temperature gradient from the measurement station to the basin average hypsometric elevation (in °C d)

$S$ = the ratio of the snow-covered area to the total area

$P$ = precipitation contributing to runoff (in cm). The parameter $Tcr$ determines whether this precipitation is rainfall and turns into runoff immediately or it is snow and should be turned into runoff with a delay (once the melting condition is right)

$A$ = the area of the basin (in $km^2$)

$k$ = recession coefficient, which indicates how much the discharge declines in the absence of rainfall or snowmelt and is given by:

$$k = (Q_{n+1})/Q_n \qquad (3)$$

where $n$ and $n + 1$ are two consecutive days in a recession period.

$n$ = the number of days in discharge calculations

Since the temperature decreases with the increase in altitude, the probability of snowfall increases. If the basin elevation range is greater than 500 m, it is best to divide it into multiple zones, each with an elevation range of 500 m. In order to implement the SRM model, WIN SRM software was used, which can be run under Windows.

### 2.3.2. Input Data

One of the inputs of the SRM is the daily temperature. This model can estimate the daily runoff of a basin based on a mean temperature as well as maximum and minimum temperatures. The next input is daily precipitation. Precipitation at higher elevations can accelerate snowmelt, leading to sharper runoff peaks. When rainfall data are available for lower elevations, they will result in underestimation without an adjustment. Thus, the rainfall data must be extrapolated from the measurement station to the basin's average hypsometric elevation [27].

In this study, the rainfall data were adjusted by 3% per 100 m change in the elevation. It should be noted that the increase in the rainfall data does not continue indefinitely and stops at a certain height. The next input is the SCA, which is defined as the ratio of the area covered by snow to the basin's total area. It can be estimated based on ground observations, aerial photographs, or satellite imagery. In this study, satellite images were used for this purpose.

### 2.3.3. Model Input Parameters

The model includes parameters determined and entered manually in the absence of sufficient data. To calculate the parameters in the region, according to previous research, the physical characteristics of the basin, the opinions of hydrology experts, and physical relationships were considered. Two of these parameters are the snow runoff coefficient ($C_s$) and rainfall runoff coefficient ($C_r$), which is the ratio of total precipitation (rain and snow) to the measured runoff. At the beginning of the snowmelt season, $C_r$ tends to be

high because losses are limited to evaporation from the snow surface, especially at higher elevations. However, as plants grow, the loss due to evapotranspiration and interception increases, resulting in a decrease in the coefficient. The runoff coefficient tends to be lower in low-flow basins, especially at lower elevations. The next parameter is the degree-day factor, which is the ratio of snowmelt depth in centimeters to the number of degree-days, as formulated in Equation (4).

$$M = a \cdot T \tag{4}$$

$$a = 1.1 \, \rho_s / \rho_w \tag{5}$$

where:

$a$ = degree-day factor (cm.$°C^{-1}d^{-1}$)

$\rho_s$ = snow density

$\rho_w$ = water density

Another parameter of the model is the temperature gradient (lapse rate). In cases where the study area has multiple temperature measurement stations at different elevations, it can be determined from the measurements. However, this is not the case in many areas. Alternatively, the temperature gradient can be estimated based on similar basins or meteorological conditions. In the absence of suitable measurements, it is common to set it to 0.65 °C per 100 m change in elevation [26]. Precipitation will be in the form of snowfall if the temperature is lower than the critical temperature. Otherwise, it will be in the form of rainfall. When precipitation is in the form of snowfall, its effect on the runoff should be studied with a delay. According to previous studies, the critical temperature ranges from +3 °C in April to +0.75 °C in July [28]. Other model parameters include rainfall contributing area, recession coefficient, and time lag. Given the time interval between the center of precipitation and the peak time of the hydrograph, a lag parameter must also be manually set by the user. This parameter can be estimated using the snowmelt hydrograph of previous years. In the studies of the World Meteorological Organization, different values have been used for this parameter. However, a suitable value for the time lag can be obtained by classifying the basin based on the size and physical characteristics and averaging the lag values calculated for each group. In this study, first, the model was executed with the described parameters set to recommended values. Moreover, sensitivity analysis was performed for the parameters [29]. Then, based on the initial results, the parameter values were modified within a certain range to increase the model's accuracy.

2.3.4. Model Accuracy Evaluation

The output of SRM is in the form of observational and computational hydrographs. Therefore, the accuracy of the model and its results can be evaluated by comparing these two hydrographs. The accuracy evaluation was performed using three metrics: the coefficient of determination ($R^2$), the volumetric difference ($D_v$), and the Nash–Sutcliffe coefficient. In order to evaluate the model accuracy, the results from 2013 to 2016 were used as the calibration period. For this purpose, the runoff data obtained from the existing hydrometric station in the area was used, and the results of 2017 and 2018 were used as the validation period.

The coefficient of determination was obtained using Equation (6) [30].

$$R^2 = 1 - \frac{\sum_{i=1}^{m}\left(Qi - \acute{Q}i\right)^2}{\sum_{i=1}^{m}\left(Qi - \overline{Q}\right)^2} \tag{6}$$

$$NSE = 1 - \frac{\sum_{i=1}^{m}\left(Si - Oi\right)^2}{\sum_{i=1}^{m}\left(Oi - \overline{O}\right)^2} \tag{7}$$

where:

$Qi$ = measured daily discharge

$Q\,'i$ = estimated daily discharge

$(\overline{Qi})$ = mean measured discharge in the target year or melting season

$m$ = number of measurement days

The formula of the Nash–Sutcliffe coefficient is given in Equation (7). It varies between 0 and 1, with values greater than 0.75 indicating good simulation, values between 0.36 and 0.75 indicating acceptable simulation, and values less than 0.36 indicating unacceptable simulation results [31].

The volumetric difference between measured and estimated runoffs was obtained using Equation (8).

$$Dv\ (\%) = \frac{V_R - \acute{V}_R}{V_R} \cdot 100 \tag{8}$$

$V_R$ = measured annual or seasonal runoff volume

$V\,'_R$ = estimated annual or seasonal runoff volume

The error in annual peak discharge estimations was calculated using Equation (9):

$$Ep = \frac{Qp, sim - Qp, obs}{Qp, obs} \tag{9}$$

Error in time to annual peak discharge was calculated using Equation (10):

$$E_{Tp} = \frac{Tp, sim - Tp, obs}{Tp, obs} \tag{10}$$

In the two last equations, $Q_{p,sim}$ and $Q_{p,obs}$ are the computational and observational peak discharges, and $T_{p,sim}$ and $T_{p,obs}$ are the time to reach computational and observational peak discharges, respectively. A lower (closer to zero) value for the two last metrics indicates a higher accuracy of the model.

## 3. Results

### 3.1. Comparison of Model Inputs

In this study, Google Earth Engine was used to estimate the SCA in daily MODIS images taken from 2013 to 2018. The obtained SCA was then divided by the total area of the catchment (zone) to obtain the fractional SCA to be given to the model as an input. In this region, the snowfall starts in early January, which coincides with the beginning of winter, and melts by the end of June, which is the end of spring. Since the goal was to examine the effect of snowmelt on flooding in a region where snowfall occurs in winter and snowmelt takes place in spring, only the first six months of each year were considered.

Daily precipitation, mean temperature ($T_{ave}$), fractional SCA, and daily runoff taken from the meteorological stations of the region were given to the model as input. The changes in the fractional SCA and mean temperature in 2013 are plotted in Figure 3. As shown, at all elevations, the SCA decreases as temperature increases. At higher elevations, the SCA is higher, and the snowmelt occurs later. Thus, in Zone A, where the SCA is the lowest, snow remains until the end of March, whereas in Zone D, where the SCA is the highest, it remains until the end of June.

Figure 4 depicts the runoff, SCA, and precipitation in the years 2013 to 2018 for the months of January to June every year. As shown, there seems to be an inverse correlation between each year's runoff and the SCA, which suggests that the snowmelt helped increase the runoff. For each year, runoff increased with the increase in rainfall. According to Figure 5, the SCA was at its peak in January and February and declined gradually, dropping to a minimum in April and May, which have the highest runoffs every year. Thus, when examining the effect of the snowmelt on flooding, more attention should be paid to the runoffs of April and May. The highest runoffs were found in 2013 and 2018 and the lowest was in 2016. The modeling results for these years are further discussed later in the paper.

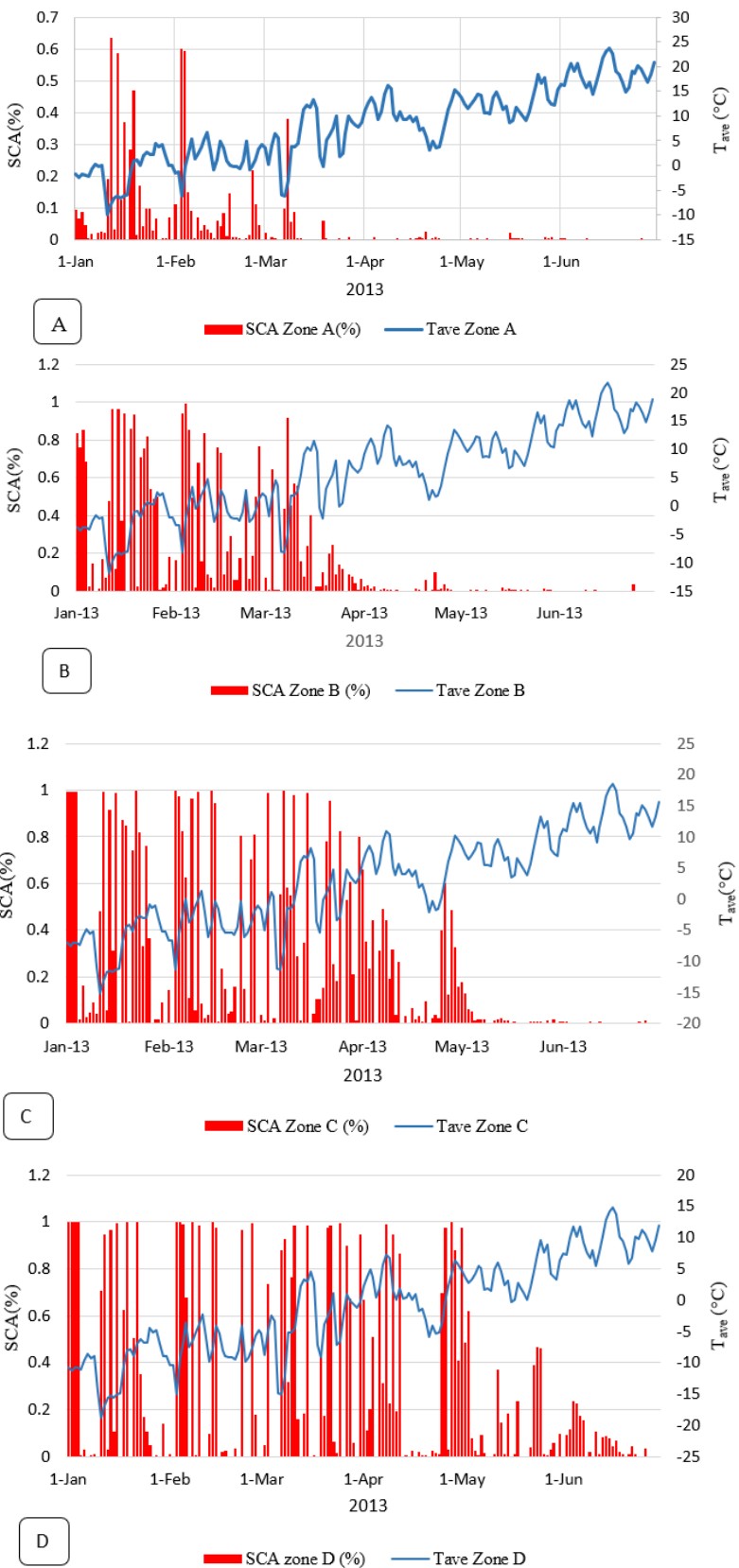

**Figure 3.** Comparison of the percentage of snow cover area and the average temperature in the (**A**) Zone A, (**B**) Zone B, (**C**) Zone C, and (**D**) Zone D in 2013.

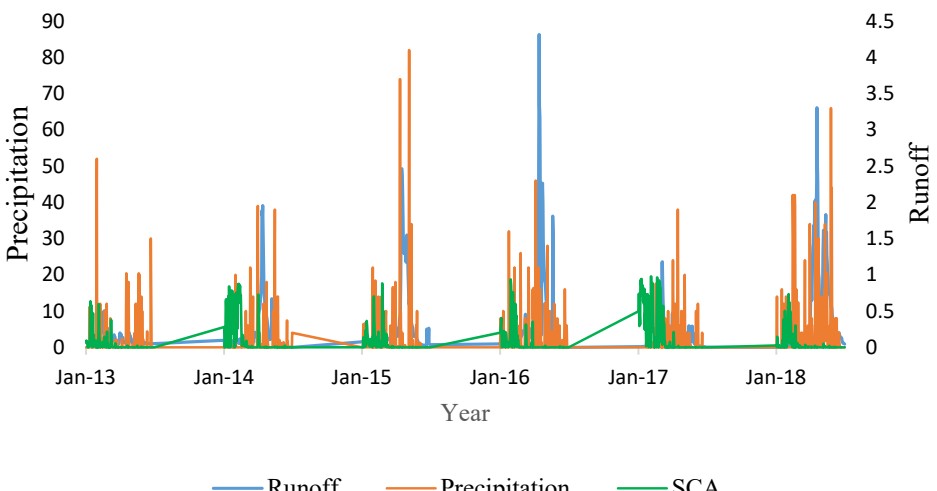

**Figure 4.** Comparison of snow, rainfall, and runoff of the study area in 2013–2018.

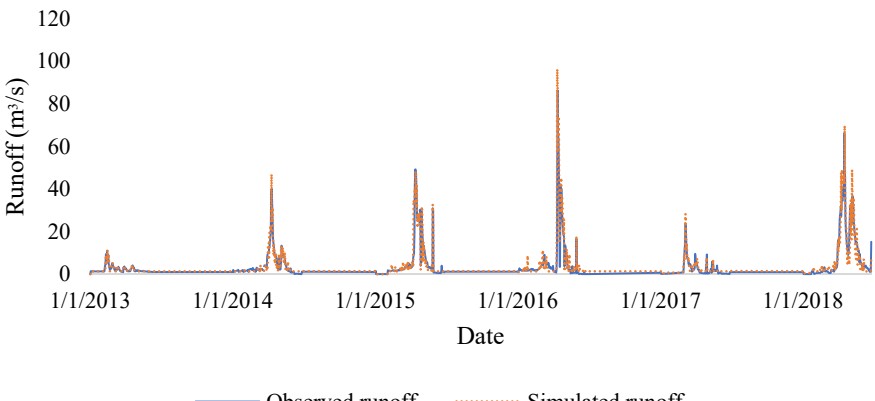

**Figure 5.** Hydrograph of the study area for the years 2013 to 2018.

### 3.2. SRM Results

In addition to the inputs described in the previous section, the SRM also requires a series of parameters to be set as explained in Section 2.3.3. As expected, the snowmelt-rainfall runoff coefficient was found to be higher in colder months because of the lack of vegetation and decreased with the growth of plants and warming of the weather. The values obtained for this coefficient ranged between 0.01 and 0.9.

The SRM with the parameter set shown in Table 3 was used to model the runoff for the years 2013 to 2018. The hydrographs obtained from the observed and simulated data for each year are compared in Figure 5. As shown, the peak runoff occurred in April 2014, 2015, 2017, and 2018, as expected, and also in February 2013 and March 2017. Overall, the comparison of the observational and simulated hydrographs demonstrated the good runoff simulation capability of the model.

**Table 3.** Variation range/value of each parameter of SRM.

| Parameter | x | y | a | $T_{crit}$ (°C) | $C_s$ | $C_s$ | $C_r$ | Lag Time |
|---|---|---|---|---|---|---|---|---|
| Value | 1.3 | 0.96 | 0.3–0.5 | 2 | 0.01–0.9 | 0.01–0.9 | 0.01–0.9 | 3 |

For the model calibration and evaluation, the period from 2013 to 2016 was considered the calibration period, and the period from 2017 to 2018 was used for the validity evaluation, which involved computing the coefficient of determination, Nash–Sutcliffe coefficient, and

volumetric difference for every year in the period. Since the goal was to examine the effect of the snowmelt on flooding, it was important to determine whether the model can accurately predict the peak runoff and the time to reach the peak. Therefore, two other factors, namely the error in the peak discharge and the error in time to the peak discharge, were also calculated. The values of the corresponding errors are given in Table 4. As shown, for most years, the Nash–Sutcliffe coefficient and the coefficient of determination were above 0.8, indicating good modeling accuracy. The volume calculations also indicated that 2013 and 2017 were dry years, which explains their lower runoff, whereas 2016 and 2018 were wet years, which explains why they had higher runoff volumes than other years. The error values computed for the peak discharge and time to peak discharge were also close to zero, indicating good simulation accuracy.

**Table 4.** Model accuracy coefficients in 2013–2018.

| | Date | Volume Observed Runoff (Million m$^3$) | Volume Simulated Runoff (Million m$^3$) | $R^2$ | $D_v$ | NSE | $E_p$ | $E_{tp}$ |
|---|---|---|---|---|---|---|---|---|
| **Calibration period** | 2013 | 32.161 | 31.948 | 0.76 | 0.66 | 0.774 | 0.03 | 0.022 |
| | 2014 | 65.295 | 62.775 | 0.84 | 3.85. | 0.856 | 0.17 | 0 |
| | 2015 | 93.766 | 95.259 | 0.86 | −1.59 | 0.86 | −0.01 | 0 |
| | 2016 | 91.772 | 104.83 | 0.82 | −14.22 | 0.82 | 0.11 | 0.022 |
| **Validation period** | 2017 | 31.372 | 36.486 | 0.85 | −16.29 | 0.83 | 0.2 | 0.022 |
| | 2018 | 136.141 | 137.15 | 0.85 | −0.74 | 0.85 | 0.05 | 0.022 |

The model outputs were also used to estimate the total runoff separately for each elevation zone and the contribution of snowmelt and rainfall to the total runoff in each zone for each year. The mean values obtained for each year are plotted in Figure 6. As shown, the total runoff is higher at higher elevation ranges. In Zone A, the total runoff is 30.7 cm, of which 58% is related to the rainfall and 42% to the snowmelt, indicating a greater contribution of rainfall to runoff generation. In Zone B, the total runoff is 36.14 cm, of which the share of rainfall is 45% and the share of snowmelt is 55%. In Zone C, the total runoff is 43.29 cm, of which 34% is related to rainfall and 66% to snow. In Zone D, the total runoff is 44.5 cm, of which the share of rainfall is 22% and the share of snow is 78%. Therefore, snowmelt has played a greater role in runoff generation than rainfall in the last two elevation ranges. These results show that the contribution of the snowmelt to runoff increases with the elevation. Of course, in this study, the investigation was conducted based on altitude zones.

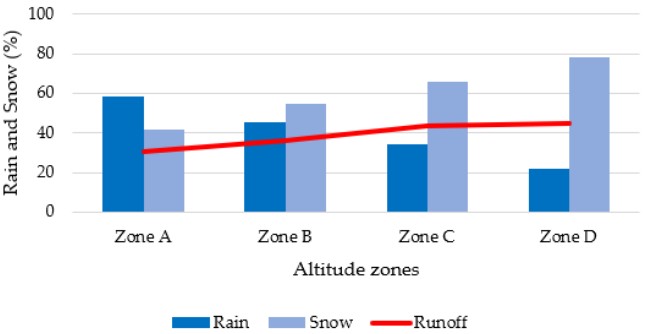

**Figure 6.** Comparison of snowmelt and rainfall contributions to runoff in four elevation zones of the study area from 2013 to 2018.

*3.3. Impact of Snowmelt on Flooding*

To determine the contribution of snowmelt to floods, we examined the total runoff in each elevation zone and then the contribution of snowmelt and rainfall to the runoff in the events (periods) during which runoff had remained high for multiple consecutive days, creating a risk of flooding. The results of this examination are presented in Figure 7.

In Zone A, rainfall contributed more to most of these events than snowmelt, except in 2014 and 2017. In Zone B, snowmelt contributed relatively more to flooding. In this zone, the snowmelt's contribution to flooding was much higher than the rainfall in February and March, occasionally going as high as 100%, such as on 4–11 February 2013. However, the opposite was true for April and May, when the rainfall contribution went as high as 100%, such as on 24–27 May 2018. Following the same trend, Zones C and D had even higher runoffs and a higher snowmelt contribution to the total runoff. Nevertheless, it is emphasized that the effect of snow on flooding in each region was considered separately.

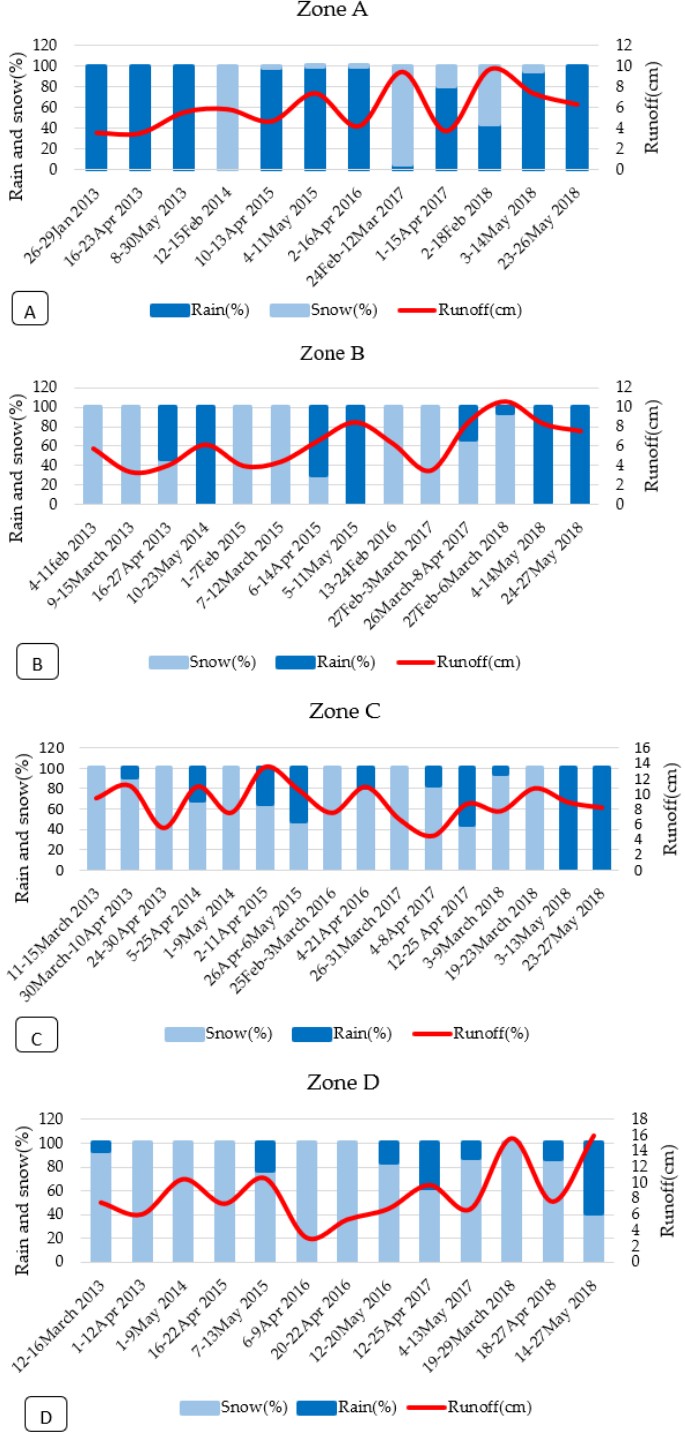

**Figure 7.** Impact of snowmelt and rainfall on flooding in (**A**) Zone A, (**B**) Zone B, (**C**) Zone C, and (**D**) Zone D.

To better illustrate the effect of snowmelt on floods, the values of all events in each zone were averaged and plotted in Figure 8. It is noted that the floods are separated in each height area. As can be seen, the rainfall and snowmelt account for, respectively, 77% and 23% of the total runoff in Zone A, reflecting a higher contribution of the rainfall to flooding. In Zone B, the contribution of the snowmelt to the total runoff has increased to 60% versus 40% for rainfall. In Zone C, the snowmelt's contribution to flooding has increased to 74%, decreasing the rainfall contribution to 26%. Finally, in Zone D, which has the highest elevation range, the contribution of the snowmelt to flooding peaked at 87%, meaning that rainfall accounts for only 13% of the flooding in the zone. Hence, the effects of the snowmelt on floods were found to increase with the increase in elevation. Thus, snowmelt appears to be an important determinant of flood volume. By comparing Figures 6 and 8, the contribution of snow melting in the total runoff in the three higher altitude regions during the melting period is almost the same in flood and non-flood periods. However, in zone A, which has more residential areas, the effect of snow on the flood state compared to the period of non-floods was less, and consequently, rainfall had more impact. This may be due to the lower permeability of residential areas and faster runoff due to heavy rains in these areas. Nonetheless, this requires a more detailed investigation of various factors in the region, which can be explored in future studies.

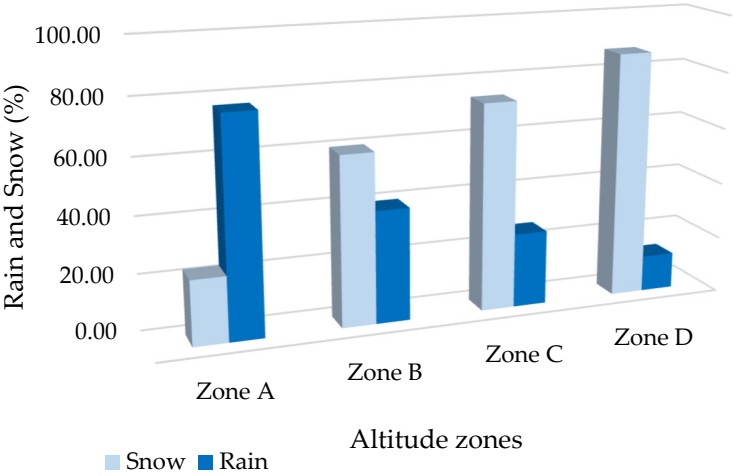

**Figure 8.** Comparison of the impact of snowmelt and rainfall on flooding in the four elevation zones from 2013 to 2018.

## 4. Discussion

The most important parameter used in the snowmelt runoff model is the fractional SCA. In the past, the common practice for estimating this parameter was to acquire, correct, and process satellite images one by one, which could be very time-consuming and was very difficult to use for longer periods [32–34]. It was common to use eight-day MODIS images, which are less accurate than daily images [35,36]. However, with the advent of Google Earth Engine, it became much simpler to process and work with satellite images. Nevertheless, not much research has been done on the effects of snowmelt on flood volume. In most studies, the effect of snow on the flooding of the whole region is considered as a whole and the effect of height is ignored [13,37]. In most of the works that have been carried out, the effects of rainfall on the snow and the faster melting of snow during floods have been considered [38–40]. In this study, in addition to examining high-altitude areas separately, the impact of the runoff caused by melting snow was also examined on floods. However, it is still not possible to consider all factors affecting runoff and floods in one study, and the results obtained have uncertainties. For example, factors such as the type of soil in the area, the slope, the amount of sunlight, land use, and land cover can certainly influence the floods caused in the area, and the impacts of each of these factors have been studied in the literature [41–44]. The calculation of the uncertainty has been investigated [45–47]. In

this study, only the effects of altitude and factors of temperature, precipitation, and snow area were considered. As explained in the results section, the snow influenced the runoff and flooding in the region, even in low-altitude areas. Obviously, neglecting snow in flood modeling will result in wrong estimates and presumptions, making it difficult to adopt the right precautionary measures to deal with floods. Even at lower elevations, where residential buildings are located, the impact of snowmelt on floods tends to be significant enough to be factored into calculations.

## 5. Conclusions

This study investigated the effect of snowmelt on flooding. Thus, it was decided to use a catchment basin encompassing both high-elevation and low-elevation areas to explore the effect of snowmelt at a range of elevations so that the results can be applied to other areas. For this investigation, the snowmelt runoff model (SRM) was used to model runoff. First, the study area was divided into four zones (i.e., A, B, C, and D) based on elevation, with Zone A having the lowest elevation range and D having the highest. Then, Google Earth Engine was utilized to estimate the snow-covered area (SCA) of each zone in the first six months of the years 2013–2018 based on the corresponding MODIS images. The results showed that the SCA decreases with the increase in temperature and increases with the increase in elevation. It was also found that runoff increases as the SCA decreases. Temperature, precipitation, and SCA data were then used as inputs for the SRM to model runoff for the years 2013–2018, and the hydrograph related to each year was obtained. The hydrographs were then compared with those plotted based on observed runoffs. The accuracy evaluation, based on the coefficient of determination, volume difference, and Nash–Sutcliffe coefficient, indicates good modeling accuracy. The error values obtained for the peak discharge and time to the peak discharge were also close to zero, demonstrating a good accuracy of the simulation. Furthermore, the contribution of snowmelt and rainfall to runoff in each elevation zone was examined separately. The examination showed that the rainfall had contributed more to flooding than snowmelt in Zone A, accounting for 58% of the total runoff. Moreover, the contribution of the snowmelt increased with elevation, ultimately reaching 78% (versus 22% for rainfall) in Zone D. The volume of runoff also increased with an increase in elevation, from 33 cm in Zone A to 44 cm in Zone D. Finally, to investigate the effect of snowmelt on flooding, the periods during which runoff remained high for multiple consecutive days, creating a risk of flooding, were identified and determined. This investigation showed that at lower elevations, snowmelt on average accounted for only 23% of the total runoff, while this ratio increased with elevation, ultimately reaching as high as 87% in Zone D. According to the results obtained in this study, snow plays a greater role in runoff and floods in mountainous areas that have more precipitation at heights above 2000 m above sea level. Ignoring the impact of snowmelt in flood modeling and flood volume estimation will result in erroneous calculations and an underestimation of flood risk, especially for high-elevation areas. Therefore, this factor needs to receive attention.

**Author Contributions:** Conceptualization, M.R.G., M.S. and M.N.; methodology, M.R.G., M.S. and M.N.; software, M.R.G., M.S. and M.N.; validation, M.R.G. and M.N.; formal analysis, M.S.; investigation, M.R.G.; resources, M.S. and M.N.; data curation, M.S.; writing—original draft preparation, M.S.; writing—review and editing, M.R.G. and M.N.; visualization, M.N.; supervision, M.R.G.; project administration, M.R.G. All authors have read and agreed to the published version of the manuscript.

**Funding:** This research received no external funding.

**Data Availability Statement:** The data that support the findings of this study are available from the corresponding author upon reasonable request.

**Conflicts of Interest:** The authors declare no conflict of interest.

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
