# Peer review of "Evaluation of Snowmelt Impacts on Flood Flows Based on Remote Sensing Using SRM Model"

_water, doi:10.3390/w15091650_

Round 1
Reviewer 1 Report
This manuscript is intended to estimate runoff from snow cover and resultant floods. The topic is of interest to the readers of the journal. The descriptions of input data and methods are clear. The presentation of results is comprehensive. The manuscript is acceptable for publication, assuming that the authors can properly address the following issues:
(1) Floods typically refer to inundations. The results from the manuscript include runoff caused by snow melting, but there is no discussion about inundation. What do the authors mean by floods?
(2) What is the basis for assuming an increase of 3% precipitation and a decrease of 0.4 degree with an increase of 100 m elevation?
(3) There is a lack of discussions about accuracy and uncertainties (e.g., daily runoffs in line 138; geometric and radiometric corrections in line 176-177; runoff values in lines 405 and 407-409)
(4) Some descriptions in the "Materials and Methods" are too general (e.g., lines 247-255). Are they relevant to your studied area?
(5) There are mix-ups between methodology and results. For example, the contents of lines 322-327 and 364-369 are not results.
(6) Isn't "calibration" and "validation" the same thing? (line 387 and Table 4)
Other comments:
What are the units of 0.73 in line 49?
What do "09" and "08" mean in figure 1?
Are the "similar works" in line 143 pertinent to your studied area?
Delete 110% in Figure 2b.
Add labels to the multiple panels of Figures 3 and 8.
Does figure 4 present just time series or really "Comparison"? Can different variables (runoff, precipitation and SCA) be compared?
The labels in figure 5 are incorrect.
Author Response
(1) Floods typically refer to inundations. The results from the manuscript include runoff caused by snow melting, but there is no discussion about inundation. What do the authors mean by floods?
Reply: We appreciate the reviewer's valuable comments. As described in Section 3.3, to determine the contribution of snowmelt to floods, we examined the total runoff in each elevation zone and then the contribution of snowmelt and rainfall to this runoff in the events (periods) during which runoff had remained high for multiple consecutive days, creating a risk of flooding and inundations. The results are also shown in Figure 7.
(2) What is the basis for assuming an increase of 3% precipitation and a decrease of 0.4 degree with an increase of 100 m elevation?
Reply: We appreciate the reviewer's valuable comments. These values ​​are given in the SRM model guide and it is suggested that if the required data for observations at higher altitudes in the study area is not available, these values ​​should be used to estimate temperature and precipitation at higher altitudes.
(3) There is a lack of discussions about accuracy and uncertainties (e.g., daily runoffs in line 138; geometric and radiometric corrections in line 176-177; runoff values in lines 405 and 407-409)
Reply: Authors appreciate reviewer’s suggestions. Added explanation about uncertainty in discussion section:
However, it is still not possible to consider all factors in runoff and flood in one study, and the results obtained have uncertainties. For example, factors such as the type of soil in the area, the slope, the amount of sunlight, land use and land cover can certainly have an effect on the flood caused in the area, and considering each of these factors can be a study work be done widely. Even the calculation of the uncertainty in these results can be done as a matter by researchers.
(4) Some descriptions in the "Materials and Methods" are too general (e.g., lines 247-255). Are they relevant to your studied area?
Reply: The authors appreciate reviewer’s comment. These explanations are given to familiarize the reader with the parameters of the model. At the end of this section (2.3.3. section), the explanations on how to obtain the parameters related to this study are also given.
(5) There are mix-ups between methodology and results. For example, the contents of lines 322-327 and 364-369 are not results.
Reply: We apologize for this mistake. These sections have been removed and moved to their respective locations.
(6) Isn't "calibration" and "validation" the same thing? (line 387 and Table 4)
Reply: Thanks for your valuable comments. In the calibration period, the parameters are determined by the explained methods and the accuracy of the modeling is also measured, and in the validation period, the determined values ​​are tested. In fact, it is the same train and test period.
Other comments:
What are the units of 0.73 in line 49?
Reply: Thanks to the reviewer's careful consideration. 0.73 is coefficient of determination and does not have unit.
What do "09" and "08" mean in figure 1?
Reply: Authors appreciate reviewer’s suggestions. The figure was modified.
Are the "similar works" in line 143 pertinent to your studied area?
Reply: We appreciate your valuable comment. It does not mean exactly the studied area in this work, but the work done in areas similar to this area.
Delete 110% in Figure 2b.
Reply: We appreciate the reviewer’s valuable suggestions. The figure was modified as suggested.
Add labels to the multiple panels of Figures 3 and 8.
Reply: Thanks to the reviewer's careful consideration. The figures are modified.
Does figure 4 present just time series or really "Comparison"? Can different variables (runoff, precipitation and SCA) be compared?
Reply: Thanks to the reviewer's careful consideration. The purpose of putting these three parameters together is to compare them at different times in such a way as to show the reader what effect it will have on the area's runoff when the rainfall or the ​​snow cover area increases or decreases. Also, according to this diagram, the delayed effect of snow on runoff can also be seen, whenever the snow cover has decreased over time, the runoff has increased.
The labels in figure 5 are incorrect.
Reply: Thank you for your attention. The labels were modified.

Reviewer 2 Report
Evaluation of snowmelt impacts on flood flows based on remote sensing using SRM model
It focuses on Evaluation of snowmelt impacts on flood flows based on remote sensing using SRM model. In my opinion, some details on the experiment should be presented in methodology that validates your results with filed data. The study is comprehensive and requires a large time to be read carefully and reviewed. The theoretical background has been well explained in detail, and the experiments and related models are presented. The paper is new and technically sounds. The result comparison parts are well organized and presented. The display way is good. The paper is properly organized, good literature review, suitable motivation, and a clear explanation of the results positive points to that. The methodology, results, and conclusions section need to be improved.
Other comments
Revise your abstract.
* What are the key results?
* What are the practical implications of your research (how can the results be utilized by e.g., readers, community)?
I assume that any change in the introduction section is not necessary, but one of the important tasks after publishing a study is to increase its chance to be seen by the most possible number of researchers. So, the more you cite a similar publication, the more the chance that the search engine in the publisher's website proposes you paper to the researcher. Besides that, it will also complete your introduction section. Another advantage, it rises new ideas for the researchers by combining various methods or resolving drawbacks of one seen paper by reading the similar one or extending the methodology to a fully automatic one. So, based on these points, I would like to ask to cite to the similar latest publication. I suggest following paper for your guideline
Ahmad, N.; Khan, S.; Ehsan, M.; Rehman, F.U.; Al-Shuhail, A. Estimating the Total Volume of Running Water Bodies Using Geographic Information System (GIS): A Case Study of Peshawar Basin (Pakistan). Sustainability 2022, 14, 3754. https://doi.org/10.3390/su14073754
Manzoor Z, Ehsan M, Khan MB, Manzoor A, Akhter MM, Sohail MT,Hussain A, Shafi A, Abu-Alam T and Abioui M (2022), Floods and flood management and its socio-economic impact on Pakistan: A review of the empirical literature.
Front. Environ. Sci. 10: 1021862.doi: 10.3389/fenvs.2022.1021862
These are the following question that needs to address from your end.
1. What are your suggestions to improve the flood flows management and quick response systems in your study area using RS and SRM model?
2. It is a serious issue of debate. I expect some appreciable and serious contributions from this platform. I advise you to revise it accordingly and provide some useful insights that are helpful to your study area and the international community.
Author Response
The methodology, results, and conclusions section need to be improved.
Reply: The authors appreciate reviewer’s comment. We tried to improve them by making changes in these sections.
Other comments
Revise your abstract.
* What are the key results?
Reply: We appreciate the reviewer's valuable comments. The key results are presented in the following:
As the height increases, the effect of snow on runoff and flooding increases, so the effect of snow, especially in high areas, should not be ignored.
* What are the practical implications of your research (how can the results be utilized by e.g., readers, community)?
Reply: The authors appreciate reviewer’s comment.
In most hydrological studies, the effect of snow is ignored due to the lack of sufficient data about snow. This study showed that snow can be very effective especially in high areas and should be given more attention in future studies.
I assume that any change in the introduction section is not necessary, but one of the important tasks after publishing a study is to increase its chance to be seen by the most possible number of researchers. So, the more you cite a similar publication, the more the chance that the search engine in the publisher's website proposes your paper to the researcher. Besides that, it will also complete your introduction section. Another advantage, it rises new ideas for the researchers by combining various methods or resolving drawbacks of one seen paper by reading the similar one or extending the methodology to a fully automatic one. So, based on these points, I would like to ask to cite to the similar latest publication. I suggest following paper for your guideline
Ahmad, N.; Khan, S.; Ehsan, M.; Rehman, F.U.; Al-Shuhail, A. Estimating the Total Volume of Running Water Bodies Using Geographic Information System (GIS): A Case Study of Peshawar Basin (Pakistan). Sustainability 2022, 14, 3754. https://doi.org/10.3390/su14073754
Manzoor Z, Ehsan M, Khan MB, Manzoor A, Akhter MM, Sohail MT,Hussain A, Shafi A, Abu-Alam T and Abioui M (2022), Floods and flood management and its socio-economic impact on Pakistan: A review of the empirical literature.
Front. Environ. Sci. 10: 1021862.doi: 10.3389/fenvs.2022.1021862
Reply: Thanks for your valuable comments. The suggested references were added.
These are the following question that needs to address from your end.
- What are your suggestions to improve the flood flows management and quick response systems in your study area using RS and SRM model?
Reply: We appreciate the reviewer’s valuable suggestions. By modeling rainfall and floods in the past years in a region using RS and SRM model as in this study and using GIS, it is possible to map flood zoning in the study area and identify high risk areas. . Then, using the modeling results, it is possible to estimate how much runoff and flood each rainfall will cause, and use the results to manage floods in the region in the future.
- It is a serious issue of debate. I expect some appreciable and serious contributions from this platform. I advise you to revise it accordingly and provide some useful insights that are helpful to your study area and the international community.
Reply: We appreciate the reviewer's valuable comments. In addition to expanding the results of our study, we tried to make suggestions for future studies.
However, in this study, in addition to examining high altitude areas separately, the impact of runoff caused by snow melting was also examined in floods. However, it is still not possible to consider all factors in runoff and flood in one study, and the results obtained have uncertainties. For example, factors such as the type of soil in the area, the slope, the amount of sunlight, land use and land cover can certainly have an effect on the flood caused in the area, and considering each of these factors can be a study work be done widely [41-44]. Even the calculation of the uncertainty in these results can be done as a matter by researchers [45-47]. In this study, only the effect of altitude and factors of temperature, precipitation and snow cover area were considered. As explained in the results section, the snow had an effect on the runoff and flooding in the region even in low altitude areas. Without a doubt, ignoring snow in flood modeling will result in wrong estimates and presumptions, making it difficult to adopt the right precautionary measures to deal with floods. Even at lower elevations, where residential buildings are located, the impact of snowmelt on floods tends to be significant enough to be factored in calculations.

Reviewer 3 Report
The article is very interesting. However, it needs some improvements.
In the introduction, please do not write too much information about the model used.
Please write how the equations of the SRM model were implemented. Was a computer program used.
Please add more citations.
Perhaps Tave is obvious but some basic description would be useful.
No subsection discussion. Please add.
How climate change affects snow melt in higher elevations.
Author Response
1) In the introduction, please do not write too much information about the model used.
Reply: Thanks for your valuable recommendation. Additional information about the model was removed.
2) Please write how the equations of the SRM model were implemented. Was a computer program used?
Reply: We appreciate your valuable comment. In order to implement the SRM model, WIN SRM software was used, which can be run under Windows.
3) Please add more citations.
Reply: We are extremely grateful for your attention. More citations were added to the manuscript.
Ahmad, N.; Khan, S.; Ehsan, M.; Rehman, F.U.; Al-Shuhail, A. Estimating the total volume of running water bodies using geographic information system (GIS): a case study of Peshawar Basin (Pakistan). Sustainability 2022, 14, 3754. https://doi.org/10.3390/su14073754
Manzoor, Z.; Ehsan, M.; Khan, M.B.; Manzoor, A.; Akhter, M.M.; Sohail, M.T.; Hussain, A.; Shafi, A.; Abu-Alam, T.; Abioui, M. Floods and flood management and its socio-economic impact on Pakistan: A review of the empirical literature. Frontiers in Environmental Science 2022, 10, 2480. https://doi.org/10.3389/fenvs.2022.1021862
Idowu, D.; Zhou, W. Land use and land cover change assessment in the context of flood hazard in Lagos State, Nigeria. Water 2021, 13, 1105. https://doi.org/10.3390/w13081105
Massari, C.; Camici, S.; Ciabatta, L.; Brocca, L. Exploiting satellite-based surface soil moisture for flood forecasting in the Mediterranean area: State update versus rainfall correction. Remote Sensing 2018, 10, 292. https://doi.org/10.3390/rs10020292
Gude, V.; Corns, S.; Long, S. Flood prediction and uncertainty estimation using deep learning. Water 2020, 12, 884, doi:https://doi.org/10.3390/w12030884. https://doi.org/10.3390/w12030884
Fenner, R.; O’Donnell, E.; Ahilan, S.; Dawson, D.; Kapetas, L.; Krivtsov, V.; Ncube, S.; Vercruysse, K. Achieving urban flood resilience in an uncertain future. Water 2019, 11, 1082, doi:https://doi.org/10.3390/w11051082. https://doi.org/10.3390/w11051082
Stephens, T.; Bledsoe, B. Simplified Uncertainty Bounding: An Approach for Estimating Flood Hazard Uncertainty. Water 2022, 14, 1618. https://doi.org/10.3390/w14101618
4) Perhaps Tave is obvious but some basic description would be useful.
Reply: Thank you for your attention. Basic description was added.
5) No subsection discussion. Please add.
Reply: We appreciate the reviewer’s valuable comments. The subsection discussion was added to manuscript.
6) How climate change affects snow melt in higher elevations.
Reply: Authors appreciate reviewer’s suggestions. Climate change affects snow cover and runoff. In another study conducted by the same authors, the effect of climate change on snow cover and runoff has been studied, and the manuscript has been revised and is under review.

Reviewer 4 Report
The article "Evaluation of snowmelt impacts on flood flows based on remote sensing using SRM model," written by authors Mohammad Reza Goodarzi, Maryam Sabaghzadeh, and Majid Niazkar, representing scientific institutions: Department of Civil Engineering, Yazd University, Iran; Department of Civil Engineering, Yazd University, Iran; Department of Agricultural and Environmental Sciences, University of Milan, Italy, is methodical and case study.
The aim of the article's work was to evaluate snowmelt impacts on flood flows based on remote sensing using the SRM model. The research area should be specified in the title. The study area (the middle Ajichai catchment) was divided into four zones based on elevation. The SnowCovered Area (SCA) from 2013 to 2018 was estimated from daily MODIS images with the help of Google Earth Engine. Runoff in the area was then simulated using the Snowmelt Runoff Model (SRM). As a result, short periods with high runoff and the possibility of floods were identified, while the contribution of snowmelt and rainfall in the total runoff was separated.
The results showed that SCA decreases with increasing temperature and increases with increasing elevation. It was also found that runoff increases as SCA decreases. Temperature, precipitation, and SCA data were then used as inputs for SRM to model runoff for the years 2013-2018 and plot the hydrograph related to each year. These hydrographs were then compared with those plotted based on observed runoffs. It was found that the share of snowmelt and runoff volume increased with height.The periods during which runoff had remained high for multiple consecutive days and creating a risk of flooding were identified, and the contribution of snowmelt and rainfall to runoff in each elevation zone were determined in these periods.
Research results, discussion, using advanced research techniques, and conclusions are logical, at a good scientific level. Among the critical remarks / for discussion, the following should be mentioned: the lack of characteristics of the geological structure, relief, climatic conditions, and hydrological conditions.
1/ Are there permeable rocks in the bedrock, and are they tectonically fractured?
2/ What is the slope of the land in the various parts of the Ajichai catchment, and what is the exposure of the catchment to geographical directions. These are factors that stimulate the maintenance of the snow cover.
3/ What is the precipitation in the Ajichai catchment and the share of snowfall. What is the monthly distribution of average air temperatures? Were the years 2013-2017 typical in terms of the amount of precipitation and the distribution of air temperatures for the values observed over many years?
4/Providing a general hydrological balance for the Ajichai catchment would be advisable. There is no hydrological characterization of the Ajichai catchment, including flow rates in the Ajichai River - annual and monthly. Can the denser network of tributaries of the Ajichai River in the right-bank part of the catchment affect the flood dynamics?
5/ All weather stations are in zone A. How did using meteorological data from zone A affect the research results?
6/ There is no definition of floods in the research area and where they are documented/observed. Was the share of snowmelt in the distinguished zones in the total runoff during the thaw period the same during the flood and flood-free periods, or did it differ significantly?
7/ In the Conclusions, it would be worth writing about the representativeness of the obtained research results - for which climatic zones and for what heights above sea level the research results could be most effectively used.
Editorial notes on the figures.
1/ Figure 1 - the figure should be supplemented with the name of the main river and the boundaries of the catchment area. Where is Lake Urmia?
2/ Figure 4 – complete in the title of the figure where the snow, rainfall, and runoff data come from
3/ Figure 5 - complete in the title of the figure where the data presented in the hydrograph for the years 2013 to 2018 come from
4/ Figure 6 - complete in the title of the figure what time period the data relate to
5/ Figure 8- complete in the title of the figure what time period the data relate to
References
All items listed in the References are cited in the text.
The article is on a good scientific level. It requires supplementation in the natural and environmental scope, defining floods and possibly distinguishing the share of snow cover melting in flood and non-flood situations. It represents the research results for a specific natural, geological, hydrological environment and climatic conditions. The article should find a wide range of readers dealing with hydrological, snow cover and environmental research, using modern research methods and tools - including remote sensing (MODIS images), Snowmelt Runoff Model, statistical analysis.
Author Response
1) Are there permeable rocks in the bedrock, and are they tectonically fractured?
Reply: Thanks to the reviewer's careful consideration. In this study, the study area has not been investigated geologically. Although this factor can also affect the results of the work, but the purpose of this work was to model snow and flood by SRM model, in this model, other factors explained in the manuscript are taken into account, and the geological structure of the area is not considered.
2) What is the slope of the land in the various parts of the Ajichai catchment, and what is the exposure of the catchment to geographical directions. These are factors that stimulate the maintenance of the snow cover.
Reply: We appreciate the reviewer’s valuable comments. In another study conducted and published by the same authors, the effect of slope and aspect on snow has been studied.
In this study, the aim is to investigate the effect of snow on floods, and snow cover is only an input in the model, so considering the slope will not affect the results.
Goodarzi, M.R.; Sabaghzadeh, M.; Mokhtari, M.H. Impacts of aspect on snow characteristics using remote sensing from 2000 to 2020 in Ajichai-Iran. Cold Regions Science and Technology 2022, 103682, doi: https://doi.org/10.1016/j.coldregions.2022.103682.
3) What is the precipitation in the Ajichai catchment and the share of snowfall? What is the monthly distribution of average air temperatures? Were the years 2013-2017 typical in terms of the amount of precipitation and the distribution of air temperatures for the values observed over many years?
Reply: We are extremely grateful for your attention.
According to observations, the average annual rainfall is 350 mm, the highest amount of rain is in the months of March and April with 20% of rainfall throughout the year, which increases the probability of flooding. The lowest amount corresponds to the months of July and August with 0.1% of precipitation throughout the year. Also, the highest amount of snow is in January and summer months without snow.
The average annual temperature in the basin is about 7.8 °C. Also, the absolute maximum annual temperature is 30°C in July and the absolute minimum temperature is -15.8°C in February.
The values ​​of temperature and precipitation in the years 2013 to 2018 are almost within the range of the long-term average temperature and precipitation of the region.
4) Providing a general hydrological balance for the Ajichai catchment would be advisable. There is no hydrological characterization of the Ajichai catchment, including flow rates in the Ajichai River – annual and monthly. Can the denser network of tributaries of the Ajichai River in the right-bank part of the catchment affect the flood dynamics?
Reply: Thanks to the reviewer's careful consideration.
The average flow rate of Aji Chai River is about 5 m3/s annually, which is the highest in April and May and about 15 and 6m3/s, respectively. Due to the low rainfall in this season and the delay of snow melting, it is expected that snow melting will play a significant role in the runoff of this river.
Undoubtedly, the density of river branches can affect the amount of flooding in that area. In order to evaluate this, it is possible to prepare a flood zoning map of the region by using GIS and identify the areas with higher risk of flooding.
5) All weather stations are in zone A. How did using meteorological data from zone A affect the research results?
Reply: We appreciate the reviewer's valuable comments. As explained in the manuscript, the temperature and precipitation data obtained from the weather stations in the region were used and entered into the model as the temperature and precipitation input of the first altitude region. Due to the lack of observation data related to higher altitude areas and using the SRM model guide, the available observation values ​​were converted to the average hypsometric height of higher altitude areas using the temperature gradient and precipitation suggested by the model and added as input to the model.
6) There is no definition of floods in the research area and where they are documented/observed.
Was the share of snowmelt in the distinguished zones in the total runoff during the thaw period the same during the flood and flood-free periods, or did it differ significantly?
Reply: Thanks to the reviewer's careful consideration.
The studied area includes 3 cities of Sarab, Haris and Bostan Abad, floods have been reported in these areas so far. Among these floods, we can mention the floods that occurred in these areas on 6/23/2018, 4/12/2015 and 4/14/2017, which were associated with human and financial losses.
By comparing Figures 6 and 8, it can be seen that the contribution of snow melting in the total runoff in the three higher altitude regions during the melting period is almost the same in flood and non-flood periods, but in zone A, which has more residential areas, the effect of snow in the flood state compared to the period No floods are less and rain has more impact. Maybe this is due to less permeability of residential areas and faster runoff due to heavy rains in these areas. However, this requires a more detailed investigation of various factors in the region, which can be investigated by researchers in subsequent studies.
7 ) In the Conclusions, it would be worth writing about the representativeness of the obtained research results - for which climatic zones and for what heights above sea level the research results could be most effectively used.
Reply: We appreciate your valuable comment.
According to the results obtained and the height of different altitude areas considered in this study, it can be seen that cold and mountainous areas that have more precipitation at heights above 2000 meters above sea level, snow plays a greater role in runoff and floods.
Editorial notes on the figures.
1 ) Figure 1 - the figure should be supplemented with the name of the main river and the boundaries of the catchment area. Where is Lake Urmia?
Reply: Authors appreciate reviewer’s suggestions. The figure was modified and added more details.
2) Figure 4 – complete in the title of the figure where the snow, rainfall, and runoff data come from
Reply: We appreciate your valuable comment. The title of figure was modified.
3) Figure 5 - complete in the title of the figure where the data presented in the hydrograph for the years 2013 to 2018 come from
Reply: Thanks for your valuable recommendation. The title was edited as suggested.
4 ) Figure 6 - complete in the title of the figure what time period the data relate to
Reply: Thanks for your valuable comments. The title was modified as suggested.
5) Figure 8- complete in the title of the figure what time period the data relate to
References
Reply: Thanks for your valuable comments. The title was edited as suggested.

Round 2
Reviewer 3 Report
The authors corrected the publication according to the guidelines therefore I think it can be published.